

# Luminescence age calculation through Bayesian convolution of equivalent dose and dose-rate distributions: the $D_e\_D_r$ model

Norbert Mercier[1], Jean-Michel Galharret[2], Chantal Tribolo[1], Sebastian Kreutzer[3,1], Anne Philippe[2]

[1]Centre de Recherche en Physique Appliquée à l'Archéologie, Université Bordeaux-Montaigne (IRAMAT-CRP2A- UMR 5060), F-33600 Pessac, France
[2]CNRS, Laboratoire de Mathématiques Jean Leray (LMJL- UMR 6629), F-44000 Nantes, France
[3]Geography & Earth Sciences, Aberystwyth University, Llandinam Building, Penglais Campus Aberystwyth, SY23 3DB, Wales, UK

*Correspondence to*: Norbert Mercier (norbert.mercier@u-bordeaux-montaigne.fr)

**Abstract.** In nature, any mineral grain (quartz or feldspar) receives a dose-rate ($D_r$) specific to its environment. The dose-rate distributions, therefore, reflect the micro-dosimetric context of grains of similar size. If all the grains have been well bleached at deposition, this distribution corresponds, within uncertainties, to the distribution of equivalent doses ($D_e$). Their combination (convolution of the $D_e$ and $D_r$ distributions in the $D_e\_D_r$ model proposed here) allows the calculation of the true depositional age. If grains whose $D_e$ values are not representative of this age (hereafter called "outliers") are present in the $D_e$ distribution, the model allows them to be identified before the age is calculated. As the $D_e\_D_r$ approach relies only on the $D_r$ distribution, the model avoids any assumption representing the $D_e$ distribution, which is usually difficult to justify. Herein, we outline the mathematical concepts of the $D_e\_D_r$ approach (more details are given in Galharret et al., accepted) and the exploitation of this Bayesian modelling based on an R code available in the R package 'Luminescence'. We also present a series of tests using simulated $D_r$ and $D_e$ distributions with and without outliers and show that the $D_e\_D_r$ approach can be an alternative to available models for interpreting $D_e$ distributions.





## 1 Introduction


For luminescence dating of sediments, the development of equipment to perform optically stimulated luminescence (OSL) analyses at the single-grain (SG) level (Duller et al., 1999a, 1999b) has been a significant technological breakthrough, offering the possibility to produce a given sample distribution of individual equivalent doses ($D_e$). This advance has also fostered the development of statistical approaches to analyse these $D_e$ distributions (e.g., Galbraith et al., 1999; Roberts et al.,

2000; Fuchs and Lang, 2001; Lepper and McKeever, 2002; Thomsen et al., 2007; Woda and Fuchs, 2008; Cunningham and Wallinga, 2012; Cunningham et al., 2015; Guibert et al., 2017; Guérin et al., 2017). Most of these statistical models target the component comprising the grains whose deposition is relevant for the event to be dated (i.e., the target population) and calculate a (believed) representative $D_e$ value from this identified sub-population. The latest proposed model (Li et al., 2021) follows the same strategy but allows identifying outliers not representative of the depositional event for several different reasons.

Therefore, all these approaches focus only on the $D_e$ distribution and require assumptions on how the individual $D_e$ values are distributed. It is also worth recalling here that the mean environmental dose rate ($D_r$) representative for the grains constituting the selected sub-population has to be determined with confidence.

In parallel to these developments, a series of investigations approached the dose rate as a cause of dispersion of the

individual $D_e$ values. These investigations were either experimental (Kalchgruber et al., 2003; Cunningham et al., 2012) and/or numerical (Nathan et al., 2003; Mayya et al., 2006; Guérin et al., 2015). They all demonstrated that the spatial distribution of radionuclide bearing minerals such as K-feldspars, but also micas or zircons, might become driving agents dominating the $D_e$ distribution. In the literature, these micro-dosimetric effects are usually grouped and considered the source of unexplained variance (overdispersion, *ext_OD*). To a lesser extent, the measurement process of the $D_e$ values causes an additional

dispersion. This component includes a purely experimental and a more theoretical part: the first refers mainly to the reproducibility of the measurement equipment, whereas the second relates to the fact that the protocol applied to determine individual $D_e$ values is not best tailored to individual grains but represents a compromise of settings deemed optimal. The dispersion induced by these phenomena constitutes the internal overdispersion (*int_OD*) which combines quadratically with the *ext_OD*.

Different experimental approaches (Rufer and Preusser, 2009; Romanyukha et al., 2017) have been proposed for quantifying the micro-dosimetric effects, whereas Martin et al. (2015a, 2015b, 2018) and Fang et al. (2018) developed numerical sediment models to calculate the $D_r$ distribution for a given granulometric fraction. Even though such experiments and applications remain rare to date, in this contribution, we want to put forward two questions: *Does the information characterizing the $D_r$ distribution provide valuable data to calculate a luminescence age?* Furthermore, if so, *What would be*

*the way to do it?* Moreover, assuming that our contribution convincingly outlines an approach: *How does such an approach help identifying intrusive or poorly bleached grains potentially present in a $D_e$ distribution?*



## 2. Convolution of $D_e$ and $D_r$ distributions

### 2.1 Basics

Let us start with a thought experiment assuming the following setting: (1) one considers a series of grains of similar shape and size behaving similarly in terms of luminescence/dose-response, (2) these grains are perfectly bleached in the laboratory and have then no residual dose, (3) they are then mixed in a matrix rich in diverse radionuclide bearing mineral phases generating a heterogeneous flux of alpha and beta particles. One also assumes (4) that the equipment used for their future analysis is perfectly reproducible. With these conditions, we expose each grain to a specific dose

rate $D_r$ which is the sum of a common gamma- and cosmic-dose contribution and heterogeneous alpha and beta-dose rate components. Because this is a thought experiment, the $D_r$ received by each grain is precisely known, enabling us to derive the $D_r$ distribution. Unfortunately, our radionuclide concentration is low, and hence we should wait for a long time (let us say 50 ka) before measuring the $D_e$ values of the individual grains. Since this is somewhat impractical, to continue, let us time-travel and measure a massive number of grains in the future: we then obtain a $D_e$ distribution whose

shape is like the $D_r$ distribution. However, in terms of doses, the first one would be characterized by a mean $D_e$ value (in Gy) equal to 50,000 times the mean $D_r$ value (in Gy.a$^{-1}$). In other words, if the two distributions are placed within the same graph, they perfectly match only if the $D_r$ distribution is shifted by a factor of 50,000, i.e., a factor corresponding to the age.

    We now go back in time and further complicate the setting by supplementing the matrix of well-bleached grains

of interest with a series of grains having non-zero residual doses, named *outliers*. After another time travel of 50 ka back into the future and the measurement of all grains, we can superimpose the two distributions as done previously to identify the outliers. Consequently, our thought experiments show that thanks to the combination of the $D_e$ and $D_r$ distributions and without any assumption about the shape of these distributions, the depositional age can be determined even if outliers are present. The mathematical details are somewhat more cumbersome than time travelling in our thought experiments,

and hence we will outline them in the following.

### 2.2 Mathematical model

    The main idea is to combine the information from the $D_e$ and $D_r$ distributions in a Bayesian framework to detect outliers automatically, i.e. grains not representative of the target population (if there are present in the $D_e$ distribution) before discarding them and computing the depositional age.




### 2.2.1 General considerations

In real life, the number of $D_e$ values measured for a sample is not extremely large. Even in cases where thousands of grains are analysed, the low percentage of grains emitting light combined with applying a series of rejection criteria may lead to a final $D_e$ sample regrouping at best a few hundred values. In contrast, when the $D_r$ values are obtained by a numerical simulation of the sediment sample, for instance, their number is only limited by the lab resources in terms of computation power.

Another key difference between the $D_e$ and $D_r$ distributions concerns individual uncertainties: each of the $D_r$ values calculated by a numerical model has no associated uncertainty. This contrasts with the $D_e$ values since each one has an error term related to the uncertainties associated with the luminescence signal and the process of its determination (fitting and interpolation). Nevertheless, the $D_r$ distribution is not free of uncertainties: at least three terms (gamma-, cosmic- and beta dose-rates) must be considered, and at least two of them (gamma and cosmic dose rates) are characterized by a mean value and an associated error.

Furthermore, one must also consider the fact that if well-bleached grains are exposed over 50 ka to a single $D_r$ value, their analysis will not produce a series of identical $D_e$ values (as expected in a perfect world), but a distribution because of the impact of the *int_OD* as discussed earlier. Such an experiment is, in fact, like a dose-recovery test (DRT) in which bleached grains are irradiated with a known laboratory source before their $D_e$ values are measured. It also means that each $D_r$ value (written $\widetilde{D}_r$ in Eq. 1 below) must be transformed to become comparable with the measured $D_e$ values since these last values include the *int_OD* effect. For this, we use the following equation:

$$D_r = \widetilde{D}_r \left(1 + int\_OD\ \epsilon\right) \qquad (1)$$

where $D_r$ is a value comparable to any $D_e$ value, *int_OD* the standard deviation characterizing the DRT distribution, and ε a Gaussian variable with uninformative mean and standard deviation (also denoted $\mathcal{N}(0,1)$).

### 2.2.2 Mathematics underpinning the model

In this section, we reiterate the method used for detecting outliers in the frame of the hierarchical model introduced by Galharret et al. (accepted). This Bayesian method can be applied for estimating an OSL age when samples of equivalent doses and dose rates are available.

We assume that the classical relation between the equivalent dose $D_e$, the corrected dose rate $D_r$ (according to Eq. 1) and the OSL age $A$:

$$D_e = A \times D_r \qquad (2)$$





is satisfied but applies to the probability distributions. More precisely, we assume that the probability distribution of $D_e$ is equal to the probability distribution of $A \times D_r$.

To determine $A$, the first step of the process is to estimate the distribution of $D_r$ on the modified sample $\widetilde{D}_r$, as described in Eq. (1). Because of the wide variety of possible distributions, we chose a Gaussian finite mixture with an unknown number of components. This is a very flexible class of distributions, allowing to catch symmetric, asymmetric, and multimodal distributions. Note that a Gaussian finite mixture model is a weighted sum of $K$ Gaussian distributions $\sum_{k=1}^{K} \dot{p}_k \mathcal{N}\left(\dot{\mu}_k, \dot{\sigma}^2_k\right)$. All the model parameters $(K, \dot{p}_1, \ldots, \dot{p}_K, \dot{\mu}_1, \ldots, \dot{\mu}_K, \dot{\sigma}_1, \ldots, \dot{\sigma}_K)$ can be easily estimated using an expectation-maximization (EM) algorithm (Dempster et al., 1977) and the optimal value of the number of components $K$ selected according to the Bayesian Information Criterion (BIC). This method is implemented in the R package 'mclust' (see Scrucca et al. 2016 for details on statistical and numerical aspects). After fitting the mixture parameters on the $D_r$ sample, we fix their values for the rest of the modelling. According to (Eq. 2), the distribution of the $D_e$ values is also approximated by a Gaussian finite mixture model with the following parameters

$$\sum_{k=1}^{K} \dot{p}_k \mathcal{N}(A\dot{\mu}_k, A^2\dot{\sigma}^2_k)$$

The second step is to estimate $A$ considering any outliers present and the measurement errors on the $D_e$ values, which are assumed to be Gaussian with zero mean and known variance. Here, the main idea of the modelling is to associate each measured $D_e$ with individual age. We denote $a_1,\ldots,a_n$ the individual ages whose associated errors are combined in the square with those of the $D_e$ values and with the systematic errors (if given). These individual ages are related to age $A$ as follows:

$$a_i = A + \epsilon_i \qquad (3)$$

where $\varepsilon_i, \ldots, \varepsilon_n$ are independent Gaussian distributions with a zero mean. In the absence of outliers, we can assume that these errors have a common variance. The density of the prior variance is then

$$p(x) = \frac{s_0^2}{(s_0^2 + x)^2} \qquad (4)$$

(cf. Galharret et al., accepted). This probability distribution is named a Shrinkage distribution with parameter $s_0^2$. This is a usual choice of prior on variance parameter for meta-analysis models (see Spiegelhalter et al. 2004). The parameter $s_0^2$ of this distribution allows controlling the dispersion of the individual ages $a_1,\ldots,a_n$ around $A$. Note that a preliminary estimate of individual ages is necessary to get an order of magnitude of the age errors. To do that, we consider the shrinkage parameter as the harmonic mean of the variance of the individual ages. This choice ensures that the errors on individual ages (resulting in particular from measurement errors on equivalent doses) is not favoured over the dispersion of the individual ages $a_1,\ldots,a_n$, and vice versa. In other words, none is assumed to be negligible relative to the other, both types of errors having the same weight on the prior information.





At this step, we may refer to this model as a Bayesian Central Age Model (BCAM) because it can be viewed as a Bayesian version of the seminal Central Age Model (Galbraith et al., 1999) even though differences exist, the most important being the absence of any pre-defined function representing the $D_e$ distribution. However, this model is not robust to the presence of outliers. Hence, before estimating $A$, we add an additional step to detect and remove the outliers
150 if they are present in the $D_e$ distribution.

In this additional step, we adapt the BCAM model in including individual random effects. This is the same principle as applied in the event model introduced by Lanos and Philippe (2017, 2018). It amounts to the assumption that the errors $\varepsilon_i, \ldots, \varepsilon_n$ have individual variances $\sigma_1^2, \ldots, \sigma_n^2$ independently and identically distributed from the same shrinkage distribution as previously chosen for the Bayesian Central Age model. While the event model can be used to
155 estimate $A$, it suffers from a lack of precision due to the summation of individual variances. Thus, in our approach, we use the posterior distribution of individual variances $\sigma_1^2, \ldots, \sigma_n^2$ for constructing a decision rule to detect outliers. Indeed, these parameters measure the dispersion of individual ages around the central age. Therefore, if an equivalent dose is detected as an outlier, its corresponding individual age will take large values with respect to the prior information on $\sigma_1^2, \ldots, \sigma_n^2$. Thus, a $D_e$ value is identified as an outlier if the posterior distribution of its individual variance is
160 stochastically greater than its prior distribution. To do that, we use quantiles and compare the prior and posterior distributions. More precisely, we fix a probability 1-α close to 1 (for instance 1-α=0.95): if the posterior (1-α)-quantile is greater than the prior (1-α)-quantile, the associated $D_e$ is tagged as an outlier and removed from the $D_e$ sample (Fig. 1).

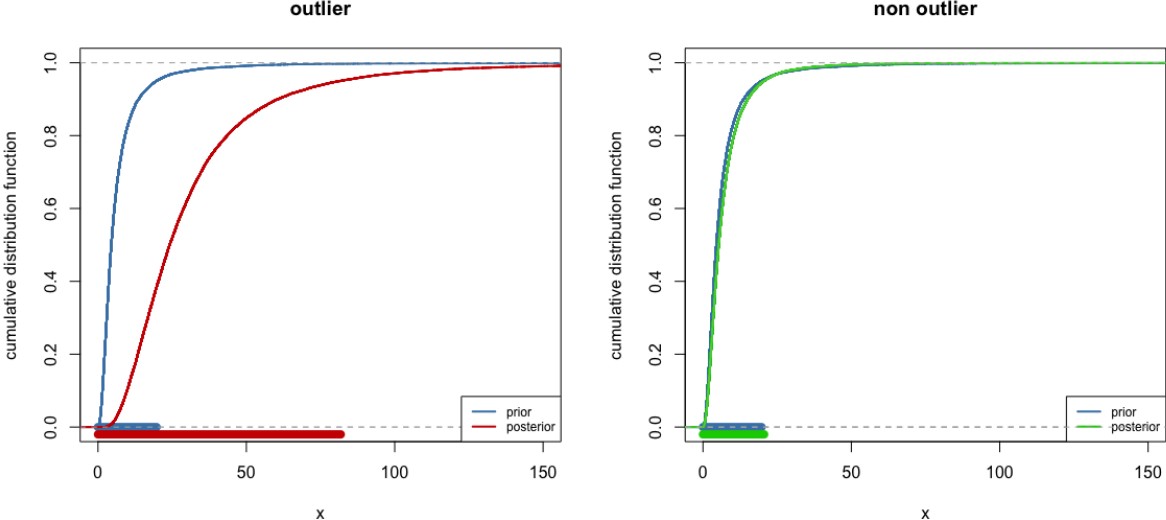

165 **Figure 1: Comparison of prior and posterior cumulative distribution functions of individual variance and their 95% credible interval (bottom horizontal lines): the corresponding equivalent dose is detected as an outlier [left] or not [right].**





When this selection is completed, the age $A$ is estimated with BCAM from the outliers' free subsample, while the posterior distributions are approximated from Markov Chain-Monte-Carlo (MCMC) samples. In practice, we use the Gibbs sampler JAGS (Plummer 2003) through the associated R (R Core Team, 2021) package 'rjags' (Plummer 2019).

### 2.2.3 Original data and structure of the model

Data input for the model are samples from the $\widetilde{D}_r$ and $D_e$ distributions. The $D_e$ distribution is a series of central values with associated errors, whereas the $\widetilde{D}_r$ distribution represents the probability of each dose-rate value. Besides, the internal over-dispersion (*int_OD*) obtained from the DRT experiment is required. This parameter allows us to compare the $D_r$ and $D_e$ distributions. Two additional values can also be given: the relative uncertainty on the average total dose rate and the relative uncertainty on the source dose rate of the equipment used for the $D_e$ measurements. However, since these parameters are considered sources of systematic errors, their effect only impacts the uncertainty associated with the obtained age.

In summary (Fig. 2), the mathematical model to combine the $D_e$ and $D_r$ distributions consists of four steps:

1. Each $\widetilde{D}_r$ value is transformed according to Eq. (1), considering the *int_OD* value,
2. the $D_r$ distribution is fitted with a weighted sum of normal (Gaussian) densities. The number of functions and their height and width are automatically adjusted to maximize the likelihood function (Fig. 3).
3. after a rough estimation of the individual ages (corresponding to the $D_e$ values divided by the mean dose-rate), the "distance" of each $D_e$ value and its uncertainty with the model is computed using an MCMC process and compared to a fixed threshold set to 5%. $D_e$ values scoring lower than 95% are considered outliers (Fig. 4),
4. finally, $D_e$ values corresponding to the identified outliers are removed from the $D_e$ sample, and the age is computed by the Bayesian Central Age Model from this new $D_e$ sample. The cumulative probability distribution of the resulting model is then compared with this new $D_e$ sample and the original data (Fig.5).



200



Figure 2: Diagram representing the different steps of the estimation method.



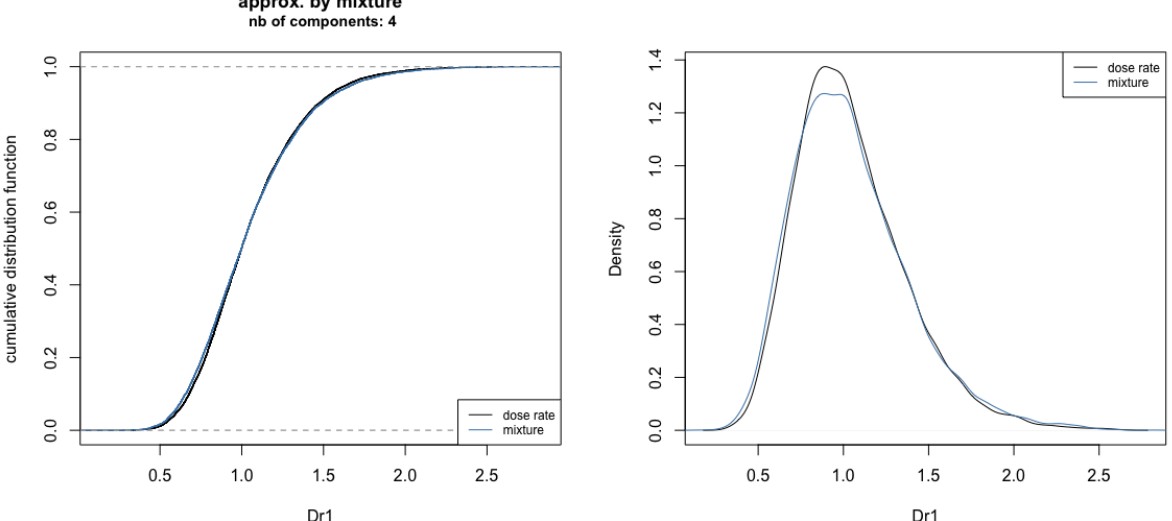

**Figure 3: Approximation of the Dr distribution with a mixture of normal (Gaussian) functions.**

225

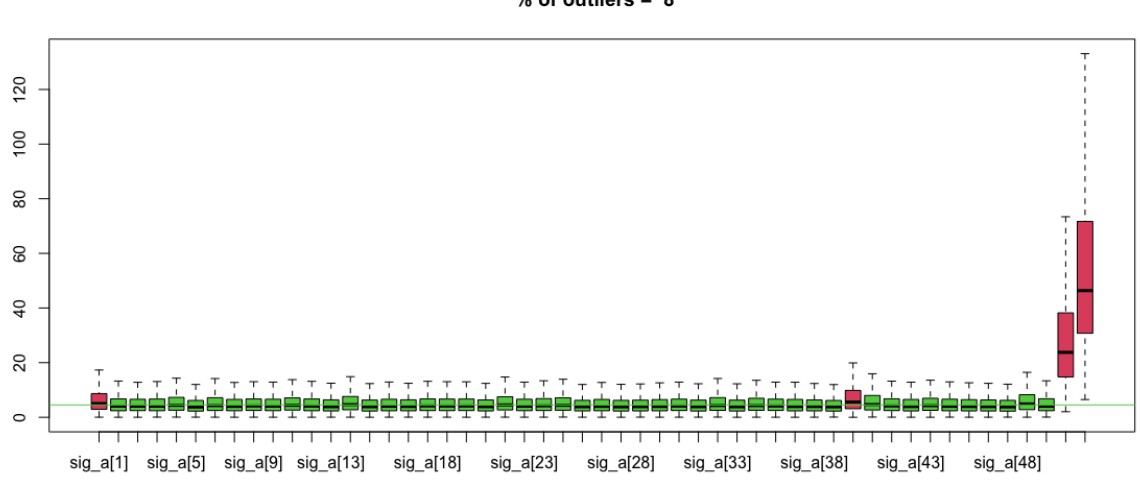

**Figure 4: Characterisation of the De values: the values in red are identified as outliers.**

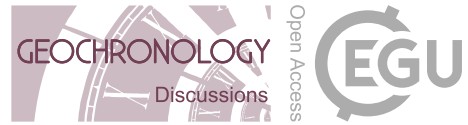



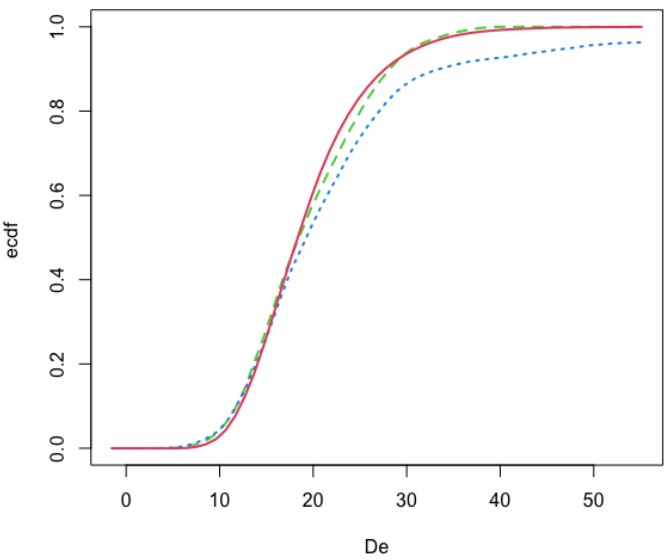

**Figure 5: Comparison of the cumulative distribution functions: A x Dr (red line), De (dotted blue line) and reduced -after removal of the outliers detected values- De (dashed green line).**

## 2.3 The implementation of the model in R

The mathematical model was implemented in R and is available in the package 'Luminescence' (Kreutzer et al., 2012) version >= 0.9.16 (Kreutzer et al., 2021) under the function name: `combine_De_Dr()`. The $D_e$ and $D_r$ distributions can be imported directly from an *Excel^{TM}* spreadsheet or CSV file or simply passed as a `data.frame` (a data object in R, comparable to a spreadsheet) imported through other formats. The other values are directly passed to the function as parameters.

The function `combine_De_Dr()` returns four plots (Supplement 1, Figs. 2–3 therein): the first two figures are related to the detection of outliers and illustrate the variation of the individual standard deviation of the posterior age distributions. The last two figures show a kernel density plot of the posterior ages, and a plot of the empirical cumulative distribution function. This last figure allows comparing the cumulative $D_e$ distributions (with or without the identified outliers), with the modelled $D_e$ distribution (A × $D_r$). We provide a simple example with R code as supplementary information (Supplement 1).



# 3. Model tests

Our tests rely on simulated numerical data (Supplement 2). Complex $D_r$ distributions were built with a series of values (at least 1,000 per series) randomly sampled from normal and/or log-normal distributions. From each obtained $D_r$ distribution, 100 values were randomly drawn and multiplied by 50 to represent individual $D_e$ values (the $D_r$ values vary around 1 Gy ka$^{-1}$, and the $D_e$ values are then around 50 and expressed in Gy). Each $D_e$ value was then associated with an uncertainty randomly sampled from a normal distribution of relative uncertainties $\mathcal{N}(0.1, 0.05)$

In case outliers were added to the initial $D_e$ distribution, their values were randomly determined from a normal or log-normal distribution, and uncertainties were defined as mentioned in the previous paragraph.

## 3.1 Tests without outliers

Table 1 lists the results of tests performed in considering four series of distributions: (1) a single normal distribution, (2) a sum of two normal distributions, (3) a single log-normal distribution and (4) a sum of two log-normal distributions. For each series, five runs were computed, giving the age obtained using the `combine_De_Dr()` function (the "apparent age"). Fig. 6 shows an example for each series.

For the four distribution types considered in these tests, the apparent age is very close to the expected age, i.e., 50 ka, indicating the efficiency of the D$_e$_D$_r$ model. It is also worth mentioning that although we had added no outlier to the initial $D_e$ distribution, a few values have been identified by the model as outliers and were then discarded before the final age was calculated. However, this is not surprising and explained by the stochastic nature of the sampling process of the $D_e$ values, which had each an associated random uncertainty.

a)                                                    b)

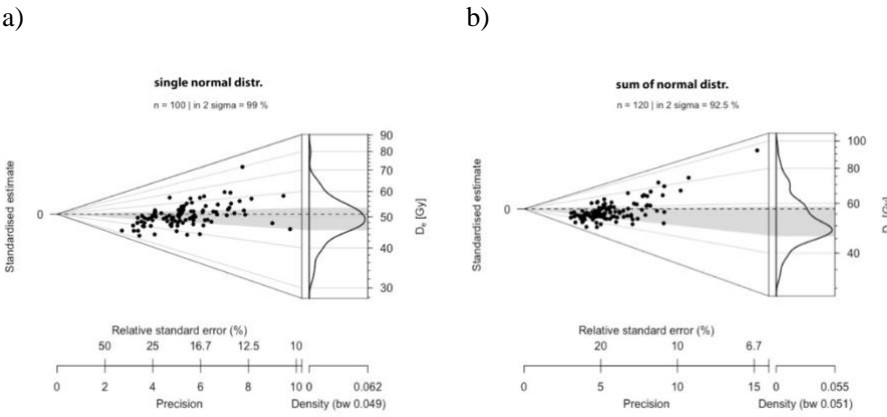



c)                                                    d)

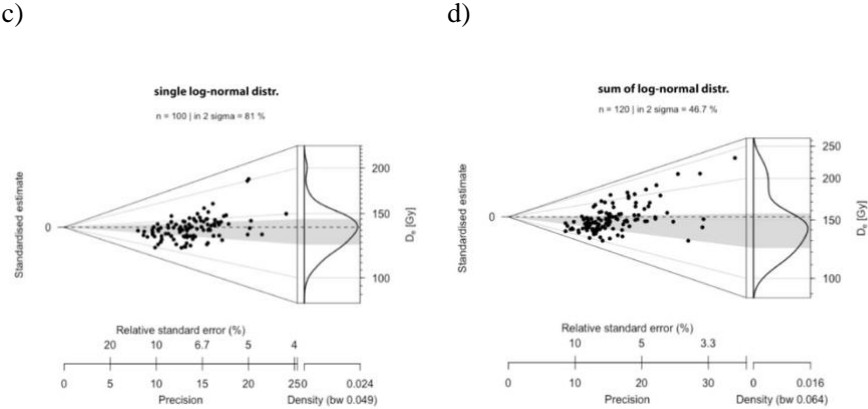


**Figure 6: Abanico plots of the De distributions (100 values) without outliers. a) single normal distribution, b) sum of two normal distributions, c) single log-normal distribution and d) sum of two log-normal distributions.**

| Dr distribution | Nb. components | Identified outliers | Apparent age (ka) | + - |
|---|---|---|---|---|
| Norm(1000, 1, 0.1) | 1 | 0 | 48.28 | 1.15 |
| | 1 | 0 | 49.71 | 1.18 |
| | 1 | 0 | 48.31 | 1.11 |
| | 1 | 0 | 50.05 | 1.14 |
| | 1 | 0 | 49.82 | 1.16 |
| | | | *49.23* | *1.15* |
| Norm(1000, 1, 0.1)+Norm(200, 1.4, 0.05) | 2 | 1 | 50.26 | 1.40 |
| | 2 | 0 | 50.50 | 1.34 |
| | 2 | 1 | 50.45 | 1.29 |
| | 2 | 0 | 49.46 | 1.39 |
| | 2 | 1 | 51.21 | 1.39 |
| | | | *50.38* | *1.36* |
| log-Norm(1000, 1, 0.1) | 1 | 1 | 48.72 | 0.78 |
| | 2 | 3 | 50.85 | 0.80 |
| | 1 | 1 | 49.87 | 0.79 |
| | 2 | 1 | 50.84 | 0.82 |
| | 2 | 4 | 49.68 | 0.81 |
| | | | *49.99* | *0.80* |
| log-Norm(1000, 1, 0.1)+log-Norm(200, 1.4, 0.05) | 2 | 2 | 49.91 | 1.03 |
| | 2 | 8 | 49.57 | 1.01 |
| | 2 | 5 | 50.66 | 1.04 |
| | 2 | 9 | 50.12 | 1.00 |
| | 2 | 9 | 49.63 | 1.03 |
| | | | *49.98* | *1.02* |

**Table 1: Results of tests without added outliers. Tests were performed with 4 series of distributions: Norm(N, m, sd) and log-Norm(n, m, sd) indicate normal and log-normal distributions, respectively, where (n) is the number of random values, (m) the**




**mean of the distribution and (sd) the standard deviation. The number of components identified by the model is given, as well as the number of points identified as outliers. Numbers in bold represent average values.**

### 3.2 Tests with outliers

Table 2 reports results for $D_e$ distributions, including 20 outliers in addition to the original 100 values. As indicated in this table, if the initial $D_e$ values were sampled from a normal distribution, the outlier values were also sampled from a normal distribution (for instance, $X_{i,k} \sim \mathcal{N}(1.3, 0.05)$ for $j := \{1,...,50\}$, $k := \{1,...,20\}$ ). Furthermore, when the 100 $D_e$ values were sampled from a log-normal distribution, the 20 outlier values were also sampled from a log-normal distribution (for instance, $X_{i,k} \sim \log\mathcal{N}(1.3, 0.05)$ for $j := \{1,...,50\}$, $k := \{1,...,20\}$).

The apparent age is slightly higher than 50 ka in all cases because a few outlier values overlap randomly with the initial $D_e$ distribution and were therefore not identified as outliers by the $D_e\_D_r$ approach. However, this over-estimation remains low (<5% of the true age), whereas outliers represent almost 17 % (20/120) of the $D_e$ values. Examples are illustrated in Fig. 7 as Abanico plots (Dietze et al., 2016).

         To test the model's performance to identify outliers when their values are close to the initial $D_e$ values, we
simulated for each type of $D_r$ distribution (normal or log-normal), outliers that followed our setting from above: $X_{i,k} \sim \mathcal{N}(1.3, 0.05)$ for $i := \{1,...,n\}$, $k := \{1,...,20\}$    and    $X_{i,k} \sim \log\mathcal{N}(1.3, 0.05)$ for $i := \{1,...,n\}$, $k := \{1,...,20\}$ where this time $n$ varied from 0 to 50 (then representing between 0 % and 33% of the initial $D_e$ distribution). The results are given in Table 3 and displayed in Fig. 8. The apparent ages increase with the percentage of outliers, but the over-estimation remains below 10% of the true age in all cases. This result is particularly interesting because these
simulations represent cases where a series of poorly bleached grains (i.e., the outliers) whose $D_e$ values are not significantly different from the mean $D_e$ have been measured in addition to well-bleached grains (initial $D_e$ values).

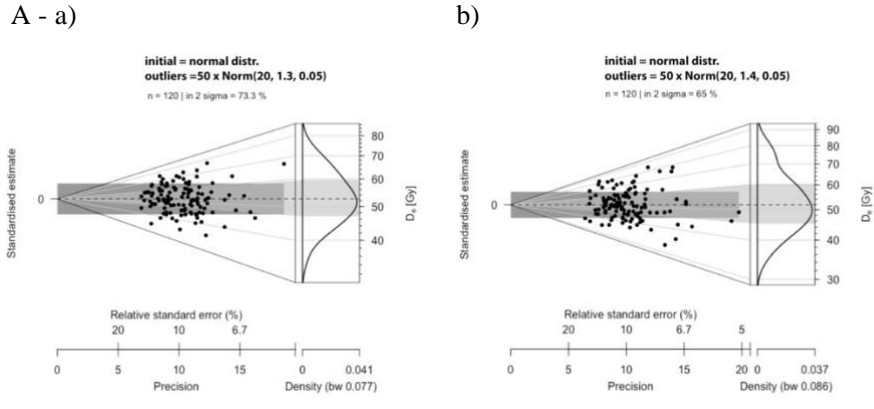


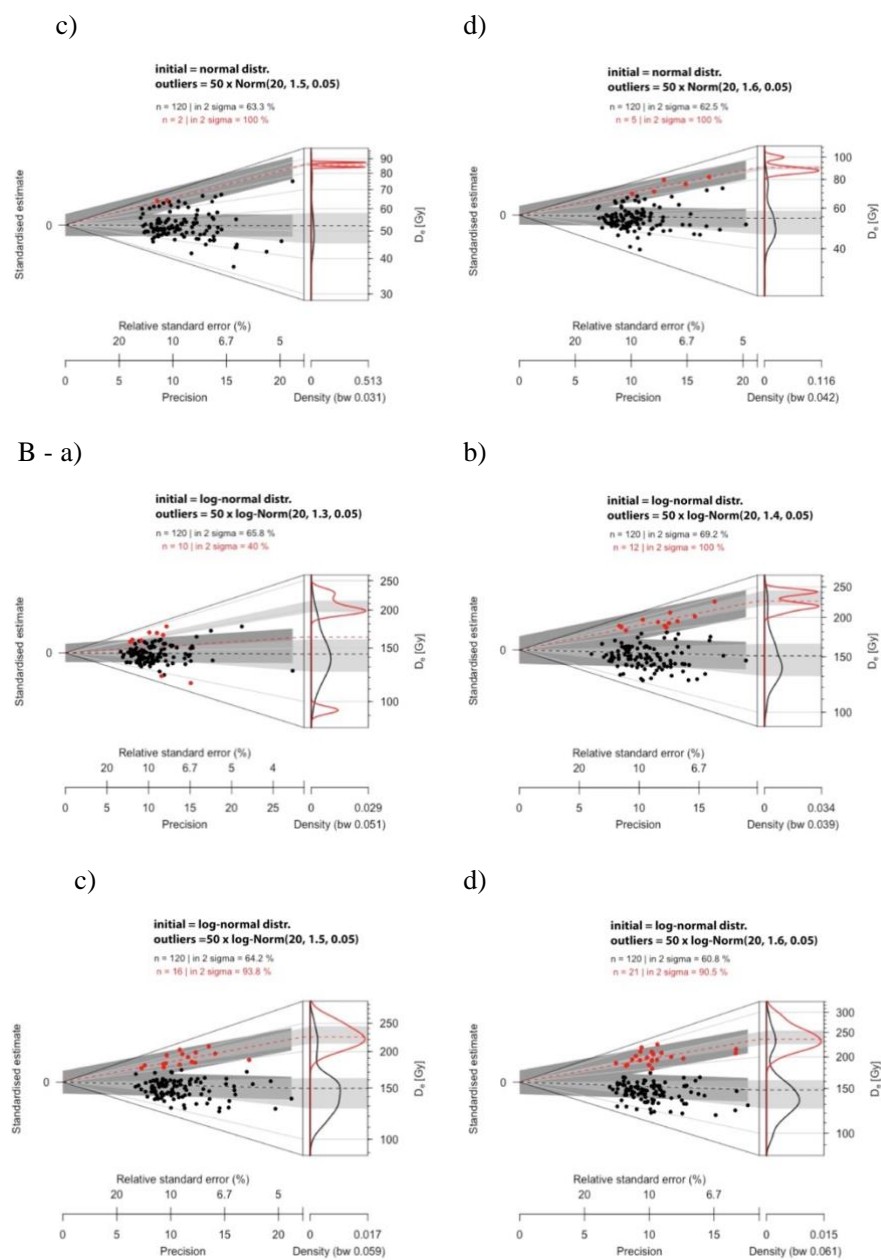

**Figure 7: Abanico plots of the De distributions (100 initial values sampled from the Dr distribution) to which 20 outlier values have been added : red dots indicate the values identified as outliers (not the values added as outliers). A- Initial De distribution = normal distribution ; B- Initial De distribution = log-normal distribution. Outlier values (20) added as follows : (A) a) 50 x Norm(20, 1.3, 0.05) ; b) 50 x Norm(20, 1.4, 0.05) ; c) 50 x Norm(20, 1.5, 0.05) ; d) 50 x Norm(20, 1.6, 0.05) and (B) a) 50 x log-Norm(20, 1.3, 0.05) ; b) 50 x log-Norm(20, 1.4, 0.05) ; c) 50 x log-Norm(20, 1.5, 0.05) ; d) 50 x log-Norm(20, 1.6, 0.05).**





| Dr distribution | Outliers distribution of De | Nb. components | Identified outliers | Apparent age (ka) | + - |
|---|---|---|---|---|---|
| Norm(1000, 1, 0.1) | 50 x Norm(20, 1.3, 0.05) | 1 | 0 | 51.37 | 1.04 |
| | | 1 | 1 | 51.35 | 1.13 |
| | | 1 | 0 | 52.01 | 1.09 |
| | | 1 | 1 | 50.93 | 1.08 |
| | | 1 | 0 | 51.04 | 1.11 |
| | | | | *51.34* | *1.09* |
| log-Norm(1000, 1, 0.1) | 50 x log-Norm(20, 1.3, 0.05) | 1 | 8 | 52.33 | 0.77 |
| | | 1 | 10 | 51.73 | 0.78 |
| | | 1 | 1 | 52.31 | 0.77 |
| | | 1 | 6 | 51.25 | 0.77 |
| | | 1 | 6 | 51.98 | 0.77 |
| | | | | *51.92* | *0.77* |
| Norm(1000, 1, 0.1) | 50 x Norm(20, 1.4, 0.05) | 1 | 0 | 52.44 | 1.12 |
| | | 1 | 1 | 51.77 | 1.07 |
| | | 1 | 2 | 52.71 | 1.07 |
| | | 1 | 1 | 51.63 | 1.09 |
| | | 1 | 4 | 51.38 | 1.14 |
| | | | | *51.99* | *1.10* |
| log-Norm(1000, 1, 0.1) | 50 x log-Norm(20, 1.4, 0.05) | 2 | 10 | 53.26 | 0.78 |
| | | 1 | 11 | 51.46 | 0.76 |
| | | 2 | 14 | 51.01 | 0.78 |
| | | 1 | 11 | 51.66 | 0.77 |
| | | 1 | 13 | 50.86 | 0.73 |
| | | | | *51.65* | *0.76* |
| Norm(1000, 1, 0.1) | 50 x Norm(20, 1.5, 0.05) | 1 | 3 | 52.87 | 1.12 |
| | | 1 | 3 | 53.70 | 1.15 |
| | | 1 | 2 | 51.37 | 1.13 |
| | | 1 | 3 | 53.64 | 1.18 |
| | | 1 | 3 | 52.37 | 1.12 |
| | | | | *52.79* | *1.14* |
| log-Norm(1000, 1, 0.1) | 50 x log-Norm(20, 1.5, 0.05) | 1 | 15 | 51.56 | 0.77 |
| | | 1 | 18 | 50.55 | 0.79 |
| | | 2 | 17 | 50.60 | 0.80 |
| | | 1 | 16 | 52.09 | 0.79 |
| | | 2 | 18 | 50.87 | 0.78 |
| | | | | *51.13* | *0.79* |
| Norm(1000, 1, 0.1) | 50 x Norm(20, 1.6, 0.05) | 1 | 2 | 52.53 | 1.11 |
| | | 1 | 6 | 51.73 | 1.14 |
| | | 1 | 7 | 51.69 | 1.14 |
| | | 1 | 7 | 51.95 | 1.11 |
| | | 1 | 10 | 52.80 | 1.15 |
| | | | | *52.14* | *1.13* |
| log-Norm(1000, 1, 0.1) | 50 x log-Norm(20, 1.6, 0.05) | 2 | 23 | 50.21 | 0.81 |
| | | 1 | 20 | 48.93 | 0.78 |
| | | 1 | 19 | 49.98 | 0.79 |
| | | 1 | 20 | 49.74 | 0.78 |
| | | 1 | 21 | 49.48 | 0.79 |
| | | | | *49.67* | *0.79* |

**Table 2: Results of tests with 20 outliers added to the original De distribution. Their values were determined following the function**
**indicated in the second column, and multiply by 50. Notice that the (m) parameter of these functions varied from 1.3 to 1.6, leading to outlier values which, on average, increased as it is observable on the Abanico plots (Fig. 7).**





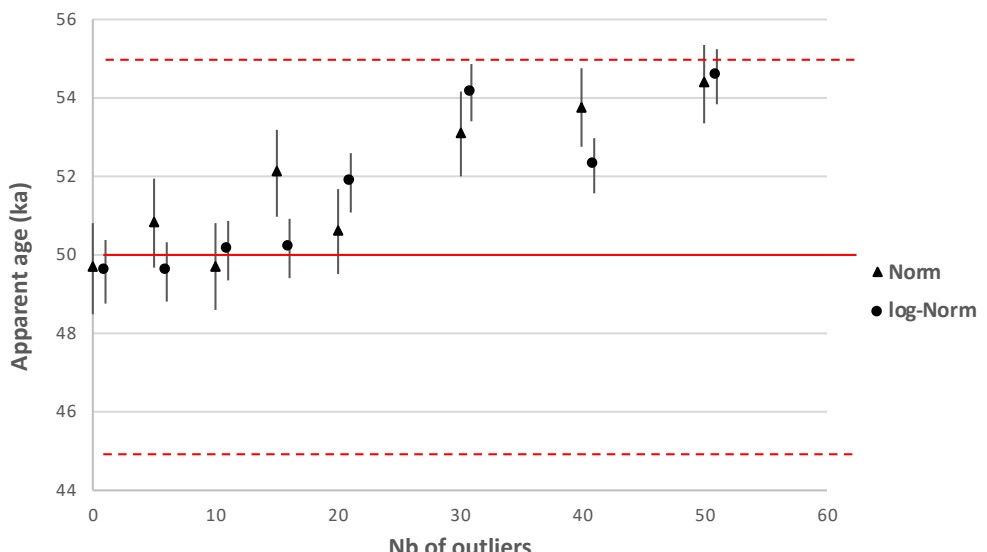

**Figure 8: Apparent age as a function of the number of outliers added to the initial De distribution (the expected age is 50 ka, indicated**
**by the red line). Norm and log-Norm represent the functions from which the initial De distributions (comprising 100 values) were**
**built. Error bars represent 95% credible intervals. Dotted lines are ±10 %.**

| Dr distribution | Outliers distribution of De | Nb. components | Identified outliers | Apparent age (ka) | + - |
|---|---|---|---|---|---|
| Norm(1000, 1, 0.1) | 50 x Norm(0, 1.3, 0.05) | 1 | 0 | 49.65 | 1.16 |
| | 50 x Norm(5, 1.3, 0.05) | 1 | 0 | 50.80 | 1.15 |
| | 50 x Norm(10, 1.3, 0.05) | 1 | 1 | 49.69 | 1.13 |
| | 50 x Norm(15, 1.3, 0.05) | 1 | 0 | 52.09 | 1.10 |
| | 50 x Norm(20, 1.3, 0.05) | 1 | 1 | 50.59 | 1.08 |
| | 50 x Norm(30, 1.3, 0.05) | 1 | 0 | 53.08 | 1.10 |
| | 50 x Norm(40, 1.3, 0.05) | 1 | 0 | 53.74 | 1.00 |
| | 50 x Norm(50, 1.3, 0.05) | 1 | 0 | 54.36 | 1.00 |
| log-Norm(1000, 1, 0.1) | 50 x log-Norm(0, 1.3, 0.05) | 2 | 1 | 49.55 | 0.80 |
| | 50 x log-Norm(5, 1.3, 0.05) | 1 | 2 | 49.56 | 0.76 |
| | 50 x log-Norm(10, 1.3, 0.05) | 1 | 7 | 50.10 | 0.76 |
| | 50 x log-Norm(15, 1.3, 0.05) | 2 | 8 | 50.15 | 0.78 |
| | 50 x log-Norm(20, 1.3, 0.05) | 1 | 6 | 51.84 | 0.76 |
| | 50 x log-Norm(30, 1.3, 0.05) | 1 | 8 | 54.13 | 0.74 |
| | 50 x log-Norm(40, 1.3, 0.05) | 1 | 12 | 52.26 | 0.71 |
| | 50 x log-Norm(50, 1.3, 0.05) | 1 | 8 | 54.53 | 0.69 |

**Table 3: For each type of Dr distribution (normal or log-normal), outliers values were added following either the function : 50 x**
**Norm(n, 1.3, 0.05), or the function : 50 x log-Norm(n, 1.3, 0.05). The number of outliers varied from 0 to 50 (then representing**
**between 0 and 33% of the initial De distribution). The age error is the 95% credible interval.**



## 4. Discussion

The previous results show that the $D_e\_D_r$ model works well for $D_e$ distributions without outliers. It also gives
satisfactory results when the $D_e$ values of the outliers are significantly different from the individual $D_e$'s composing the target population. On the other hand, the existence of values defined as outliers but very close to the target population may be assimilated by the model to the target population and thus not be identified as outliers. This is related to the fact that the $D_e\_D_r$ model is a majority rule model.

This notion of majority is vital because it sets the limits of applicability of the model. If the number of outliers
is *significantly* larger than the number of $D_e$ values representing the target population, the $D_e\_D_r$ model will combine the $D_e$ and $D_r$ distributions as best as possible so that a maximum of $D_e$ values corresponds to $(A \times D_r)$ values. A visual examination of the distributions calculated by the model (e.g., Fig. 5) is therefore indispensable, as it is for Fig. 4, which shows the identified outliers within the distribution of individual ages.

On the plus side, it is also important to recall that the De_Dr model does not require a predefined function
representing the $D_e$ distribution. However, it does require a precise determination of the $D_r$ distribution. To date, this distribution can be obtained either from numerical sediment models or obtained experimentally using nuclear detectors (e.g., Romanyukha et al., 2017). Unfortunately, at present, such experiments are scarce, but if they become more common, perhaps cases will be observed where the $D_r$ distributions do not follow locally, within the samples, a simple distribution (typically log-normal).

To date, the De_Dr model is thus the first model that allows considering the information from the equivalent doses and dose rates simultaneously, thus offering a substantial paradigm change compared to existing approaches.

## 5. Conclusion

The De_Dr model is an alternative to statistical models to determine the target population from a $D_e$ distribution. Combining the information associated with the equivalent doses and dose rates experienced by the grains during burial,
the model offers the possibility to determine the age of the target population without any predefined function representing the $D_e$ distribution.

Future work should focus on tests carried out on well-dates samples (typically cross-checked with $^{14}$C dating) to validate the De_Dr model experimentally. This would, however, necessitate first to have access to accurately and precisely determined $D_r$ distributions.




**Code and data availability.** The source code of the model is part of the R package 'Luminescence' (>= v0.9.16) and available open-access under GPL-3 licence conditions (https://CRAN.R-project.org/package=Luminescence; last accessed: 2021-09-08).


**Author contributions.** NM and CT initiated the work, wrote the first manuscript draft and an initial R script. JMG and AP developed the mathematical basis for the model. SK implemented the model in the R package 'Luminescence' and ran additional tests. All authors equally contributed to the discussion and the final manuscript write-up.

**Competing interests.** The authors declare no competing interests.

**Financial support.** This work received financial support from the LaScArBx LabEx, a programme supported by the ANR (ANR-10-LABX-52). The contribution of S. Kreutzer received funding from the European Union's Horizon 2020 research and innovation programme under the Marie Skłodowska-Curie grant agreement No 844457 (CREDit).

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
