# Peer review of "Luminescence age calculation through Bayesian convolution of equivalent dose and dose-rate distributions: the De\_Dr model"

_Geochronology, 2021_

## Referee Comment (RC1)

**Specific Comments**

Line 15: change 'this distribution corresponds' to 'this distribution is assumed to correspond'

Line 16: change 'allows the calculation' to 'would then allow calculation'

Line 18: Add ', enabling their exclusion.' after 'before the age is calculated' and 'to describe the $D_e$ distribution' after 'the $D_r$ distribution'

Line 19: change 'any assumption representing the De distribution' to 'any assumption of the shape of the De distribution' and 'is usually' to 'can be'

Lines 43-44: change 'the source' to 'a significant source'. I also think you should add one or more sentences following this one that discuss other significant sources of external overdispersion (e.g. sediment mixing or incomplete bleaching) and make a distinction between external overdispersion caused by outlier grains and external overdispersion caused by the Dr distribution inherent to the sample.

Line 62: delete 'in the laboratory' and change 'have then no residual dose' to 'have no residual dose'

Lines 66-73: The text about 'waiting a long time' and 'time-travel', etc. is distracting and unnecessary as it is already clear this is a thought experiment. I suggest replacing all these lines with a single sentence such as: 'If we wait 50 ka and measure a massive number of $D_e$ values from these grains, we would expect to obtain a $D_e$ distribution with the same shape as the $D_r$ distribution but offset by a factor of 50,000.'

Lines 74-77: The text about time travel is again distracting and unnecessary. I recommend removing such references to make the sentences more concise. For example 'If the depositional setting was further complicated by supplementing the matrix of well-bleached grains of interest with a series of grains having non-zero residual doses, then superimposition of the De and Dr distributions could potentially identify these outliers. '

Line 79-80: Delete 'time travelling in our' and add 'section' after 'following'

Lines 83-84: Change to "detect outliers (i.e. grains not representative of the target population) automatically (if they are present), before discarding them and computing the depositional age.'

Line 89: Change '$D_e$ sample regrouping at best a few hundred values' to '$D_e$ distribution comprised of at best a few hundred values'

Line 94: In your paper you state that 'each $D_r$ value calculated by a numerical model has no associated uncertainty'. I'm guessing this may be true for DosiVox or other current models (I have not used them and so have limited experience in this area) and based on reading your paper and looking at the code in the luminescence R package it seems that your $D_e\_D_r$ model doesn't currently have a means of incorporating uncertainties for values in the $D_r$ distribution. However these values certainly have uncertainties even if they are not known or calculated by current models. I might hope that in the future estimates of those uncertainties might be calculatable and then included in future versions of your method. I think it would be prudent, therefore, to modify the statement in your paper to something such as 'current numerical models of $D_r$ distributions do not report uncertainties for individual $D_r$ values'.

Line 101: Modifying dose rates based on the overdispersion of a dose recovery test is a new and strange concept and I think your paper could be improved by clarifying the reasoning for the modification. For example: 'As the $D_e\_D_r$ model relies on the shape of the $D_r$ distribution to describe

the expected shape of the $D_e$ distribution and identify outliers, the int_OD of the $D_e$ distribution (such as measured with a DRT) needs to be incorporated into the $D_r$ distribution. To do this, individual $D_r$ values from a $D_r$ distribution of a numerical model ($\sim D_r$) are transformed into internally overdispersed $D_r$ values ($D_r$) using the following equation:'

Lines 111-112: "This Bayesian method can be applied for estimating an OSL age when samples of equivalent doses and dose rates are available." In this sentence (and at several other places within the paper) the term 'sample' is used in a statistical sense to refer to a set of values from a distribution. Although this is a correct usage of the term, it may be confusing for the audience of this paper who are more accustomed to using the term 'sample' to refer to mineral grains collected from a sediment deposit. I suggest clarifying your message here and throughout the paper by retaining the word 'sample' for this second definition. For instance, this sentence could be changed to something such as "This Bayesian method can estimate an OSL age for samples with both single-grain equivalent dose values and simulated (or measured) dose rate distributions."

Lines 118-119: "estimate the distribution of $D_r$ on the modified sample $\sim D_r$, as described in Eq. (1)" I recommend modifying the text so that the term 'sample' is only used to refer to the mineral grains being dated. For example "estimate the sample's $D_r$ distribution when the internal overdispersion of the $D_e$ distribution is incorporated, as described in Eq. 1."

Line 126: change 'sample' to 'distribution'

Lines: 146-147: "we may refer to this model as a Bayesian Central Age Model (BCAM) because it can be viewed as a Bayesian version of the seminal Central Age Model (Galbraith et al., 1999)". My understanding of what makes CAM (Galbraith et al 1999) unique compared to any other weighted average of a distribution is that it a) uses the logs of the individual equivalent dose values for fitting the distribution and b) calculates a median rather than a mean for the central value. Does your BCAM also do these things?

Line 167: Change "outliers' free subsample" to "$D_e$ distribution with the outliers removed"

Line 171: Change 'Data input' to 'Input data' and 'samples' to 'values'

Lines 173-174: 'This parameter allows us to compare the $D_r$ and $D_e$ distributions'. Comparisons are possible anyways, please consider changing your text to reflect that this parameter is used to modify the $\sim D_r$ distribution to be the same shape as the expected $D_e$ distribution.

Line 187: Change 'sample' to 'distribution'

Line 188: Change 'sample' to 'distribution'

Line 189: Change 'sample' to 'distribution'

Figure 2: change '$\sim D_r$ sample' to '$\sim D_r$ distribution' and '$D_e$ sample' to '$D_e$ distribution'. Also change '$D_e$ subsample after removing outliers' to '$D_e$ distribution after removing outliers'

Line 254: change 'in considering four series of distributions' to 'using four different $D_r$ distributions'

Line 256: change 'series' to '$D_r$ distribution'. Also, I would recommend using '$D_e\_D_r$ Age' to refer to ages calculating using your model instead of 'apparent age'. So, in this instance, you might consider changing 'giving the age obtained using the combine_De_Dr() function (the "apparent age")' to something along the lines of 'and $D_e\_D_r$ Ages were calculated using the combine_De_Dr() function.'

Line 257: change 'Fig. 6 shows an example for each series' to 'Fig. 6 shows an example Abanico plot (Dietze et al., 2016) of a $D_e$ distribution for each type of simulated $D_r$ distribution'

Line 258: change 'apparent age' to '$D_e\_D_r$ Age' and 'expected age' to 'given age'

Table 1: change 'apparent age' to '$D_e\_D_r$ Age'

Table 1 caption: change '4 series of distributions' to '4 different shapes of $D_r$ distributions'. Also, I'm a bit confused what the number of components refers to. Is it the number of Gaussian distributions fitted to the Dr distribution? If so, can you please specify that to make it more clear.

Line 285: change 'apparent age' to '$D_e\_D_r$ Age'. Also please mention that the added outlier values represent grains with residual doses and are therefore greater than the target De distribution.

Line 290: change 'simulated for each type of distribution (normal or log-normal), outliers that' to 'simulated normal and log-normal distributions with outliers that'

Line 293: change 'apparent ages' to '$D_e\_D_r$ Ages'

Figure 7 and Table 2: It is confusing that the results are ordered differently in figure 7 and Table 2. I suggest re-ordering the results of Table 2 similarly to how they are presented in Figure 7.

Figure 7: The two tier lettering system of A-a-d and B-a-d is confusing. Please re-letter the subfigures as a-h and specify in the caption that a-d are based on normal distributions and e-h are based on log-normal distributions

Table 2: change 'apparent age' to '$D_e\_D_r$ Age'

Table 2 caption: 'values were determined following the function indicated in the second column and multiply by 50'. Please delete 'and multiply by 50' as the function in the second column already includes this multiplication. Also change 'as it is observable' to 'as is observable'

Figure 8: Is one of the datasets offset? Otherwise wouldn't the norm and log-Norm data for say 50 outliers plot on top of one another instead of side by side? Also could you please display one of the datasets with open symbols for easier distinguishing between them.

Line 329: delete 'The previous'

Lines 337-338: change 'as it is for Fig. 4, which shows the identified outliers within the distribution of individual ages.' to 'as is a visual examination of the outliers identified within the distribution of individual ages (e.g. Fig. 4).'

Line 343: I'm not sure I understand what you mean by 'locally, within the samples,'. Please consider deleting these words and just saying 'do not follow a simple distribution' or some other alternative wording with greater clarity.

**Technical Corrections**

Line 13: change 'any' to 'each'

Line 14: change 'have been' to 'were'

Lines 15-16: delete the parentheses and the word 'convolution'

Line 18: change 'the model' to 'this model'

Line 28: change 'a given sample distribution of individual equivalent doses ($D_e$).' to 'a distribution of individual equivalent doses ($D_e$) for a given sample.'

Line 56: change 'identifying' to 'identify'

Line 144: change 'none' to 'neither'

Line 171: Change 'Besides' to 'Additionally'

Line 242: change 'allows comparing' to 'compares'

Line 251: change 'In case' to 'In cases where'

Line 258: change 'indicating' to 'demonstrating'

Line 259: change 'although we had added no outlier' to 'although we didn't add outliers'

Line 261: change 'and explained' to 'and can be explained'

Line 262: change 'which had each' to 'which each had'

Line 280: change 'distributions, including' to 'distributions including'

Line 340: change 'representing' to 'to represent'

Line 352: change 'well-dates samples' to 'well-dated samples'

Line 353: change 'necessitate first to have access' to 'first necessitate access'

---

## Author Response (AR1)

Dear Julie,
Dear Alastair,
Dear Anonymous Reviewer

Once more, thank you for your thorough comments and suggestions, which have greatly helped us to improve our manuscript.

We have followed all suggestions and corrections proposed by reviewer 1 to render our text more concise and intelligible. We have also been more careful and precise in using certain words (sample, distribution) with the same objective.

Reviewer 1 raised the issue of the uncertainty associated with the dose rate, which we did not address in the first version of the manuscript. We have studied this issue from a mathematical point of view and have provided a detailed answer in the "Discussion". Indeed, this is an important point that was missing from the discussion.

In the discussion, we also elaborate more on the reasons for testing the effectiveness of the De_Dr approach on simulated datasets. As Reviewer 2 (Alastair) rightfully pointed out, it will be necessary to test the relevance of the De_Dr model further as soon as more reliable $D_r$ distributions of samples are available. We understand that not having this included renders a weak point in our manuscript because, with such data, many of the questions raised by Alastair might be tested. Unfortunately, such distributions are not readily available and will need significantly more resources, if not an own project dedicated to it. Here, the additional context (many degrees of freedom to consider) likely distracts from the key message of our study. Hence, intentionally, we had decided to provide this first concise manuscript, a transcription of the underlying Bayesian approach and the process for identifying outliers. It contains the model and the R code freely accessible to the luminescence community, something to further build on in the future.

On behalf of all co-authors,

Norbert Mercier